# The PELP1 Pathway and Its Importance in Cancer Treatment

**DOI:** 10.3390/biom15121729

**Published:** 2025-12-12

**Authors:** Khaled Mohamed Nassar, Panneerdoss Subbarayalu, Suryavathi Viswanadhapalli, Ratna K. Vadlamudi

**Affiliations:** 1Department of Obstetrics and Gynecology, University of Texas Health San Antonio, San Antonio, TX 78229, USA; nassark@livemail.uthscsa.edu; 2Greehey Children’s Cancer Research Institute, University of Texas Health San Antonio, San Antonio, TX 78229, USA; subbarayalu@uthscsa.edu; 3Audie L. Murphy Division, South Texas Veterans Health Care System, San Antonio, TX 78229, USA

**Keywords:** PELP1, PELP1 inhibitors, Rix complex, oncogene

## Abstract

Proline-, glutamic acid-, and leucine-rich protein 1 (PELP1) is a proto-oncogene that serves as a nuclear and cytoplasmic scaffolding protein. PELP1 plays a critical role in nuclear receptor signaling, ribosome biogenesis, chromatin modifications, cell cycle progression, non-genomic signaling, and DNA damage response. PELP1 expression is upregulated in a variety of cancers, including breast, ovarian, endometrial, prostate, and liver cancers and serves as a prognostic factor for poor survival. PELP1’s structural motifs, unique scaffolding function, and oncogenic activity make it a potential target for a range of therapeutic approaches. This review summarizes the most recent advancements in PELP1 biology, with a particular focus on the emergent oncogenic functions of PELP1 and its inhibitors for the treatment of cancer.

## 1. Introduction

Proline-, glutamic acid-, and leucine-rich protein 1 (PELP1) is a multifunctional scaffold protein that acts as a transcriptional coregulator for various nuclear receptors (NRs) and transcription factors [1,2,3]. PELP1 also serves as an important regulator/mediator of genomic and nongenomic signaling by connecting NRs to cytoplasmic kinases like Src, PI3K, and MAPK [2,4]. PELP1 is an important component of the Rix1 complex and plays a critical role in ribosome biogenesis and new protein synthesis [5,6,7,8]. PELP1 interacts with several chromatin-modifying enzymes, such as HDAC2, KDM1A, and CBP/p300, which help histone modifications at gene promoters. Recent research implicates PELP1’s role in cell cycle progression, DNA damage response, stem cell maintenance, metabolic signaling, and immune modulation. PELP1 expression is commonly deregulated in many solid cancers, establishing it as a potential therapeutic target. This review summarizes recent findings regarding the oncogenic functions of PELP1, with an emphasis on the literature published within the past decade, and evaluates therapeutic strategies designed to target this protein.

## 2. PELP1: Structure and Function

### 2.1. Overview of PELP1 Structure

PELP1 lacks known enzymatic activity, predicted to function as a scaffolding protein with a theoretical molecular weight of 120 KDa (Figure 1). Because of the presence of excess proline and glutamic acid amino acids, it runs as ~160 kDa in Western blots. PELP1 contains 10 LXXLL motifs which contribute to its interaction with NRs, including estrogen receptors (ERs), androgen receptors (ARs), and glucocorticoid receptors (GRs) [1]. In addition, PELP1 contains multiple PXXP (P refers to proline, X refers to any residue) motifs that promote its interaction with the proteins that contain the Src homology 3 (SH3) domain, such as the signaling protein c-Src [9,10]. At the C-terminus, PELP1 contains a long stretch (~70 amino acids) of an acidic and glutamic-acid region that binds to histones, facilitating PELP1 to regulate chromatin [1,11]. PELP1 has centrally localized the nuclear localization signal (KKLK) that facilitates its nuclear localization [12]. PELP1’s N-terminal region comprises two nucleolar domains, Nuc 1 (amino acids 79–160) and Nuc 2 (423–489), which help localize PELP1 in the nucleolus [13]. PELP1 contains two cysteine-rich domains interspersed with a nuclear localization signal. This area possesses the capacity to generate three zinc finger-like motifs, enabling PELP1 to facilitate protein–protein interactions and reinforce its structure through zinc coordination. PELP1 contains the RNA recognition motif, binds RNA with a preference to poly-C, co-localizes with the splicing factor SC35 at nuclear speckles, and participates in alternative splicing [14]. Collectively, these findings indicate that PELP1’s modular structure enhances its interactions with a varied array of signaling complexes and thus supports PELP1’s versatility as a signaling hub in aiding oncogenic signaling.

### 2.2. PELP1 Post-Translational Modifications

PELP1 is modified by a variety of post-translational modifications. Cyclin-dependent kinases (CDKs) phosphorylate PELP1 (Ser477 and Ser991) during cell cycle progression, and these changes are associated with enhanced activation of E2F1 target genes and ER signaling [15]. Growth factor signals promote the phosphorylation of PELP1 (Ser350, Ser415, and Ser613) through PKA, and such modifications enhance PELP1 co-activation functions via nuclear redistribution of PELP1 [16]. DNA damage response (DDR) kinases such as ATM, ATR, and DNA-PKcs phosphorylate PELP1 at serine 1033, a modification that plays a critical role in regulating DNA damage response and apoptosis [17]. PELP1 interacts with GSK3β and serves as a new substrate for GSK3β. Phosphorylation of PELP1 at Thr745 and Ser1059 by GSK3β is associated with the stability of PELP1 [4]. PELP1 undergoes SUMOylation, which is responsible for the coordination of ribosome formation through small Ubiquitin-like Modifier (SUMO)-dependent subnuclear trafficking [18]. Ubiquitination of PELP1 at K496 by Vps11/18 is sufficient to prevent its interaction with c-Src and the subsequent phosphorylation of ERα by c-Src [19]. PELP1 is a substrate of tubulin tyrosine ligase-like family member 4 (TTLL4), and PELP1 polyglutamylation affects its affinity to histone H3 and other proteins and subsequent chromatin remodeling [20]. c-Src phosphorylates PELP1 at Tyr920, indicating reciprocal regulation between the two proteins [21]. These post-translational modifications highlight the complex regulatory network that governs PELP1's diverse cellular functions (Table 1).

### 2.3. PELP1 Interaction Partners

Nuclear receptors (NRs): PELP1 functions as a coactivator of multiple NRs (Table 2). PELP1 interacts with various NRs via its ten different LXXLL motifs, including ERα [12], ERβ [23], AR [24], PR [25], GR [26], estrogen-related receptor α (ERRα) [27], and retinoid X receptor (RXR) [28]. PELP1 has also been shown to be a coregulator for nuclear transcription factors like activator protein 1 (AP1), specificity protein 1 (SP1), nuclear factor κB (NF-κB) [29], signal transducer and activator of transcription (STAT3) [30], and four-and-a-half LIM protein 2 (FHL2) [3] as illustrated in Figure 2.

Kinases: PELP1 also functions as a cytoplasmic scaffolding protein [31]. PELP1 interacts with p85 subunit of PI3K and also with EGFR in breast cancer (BCa) cells, activating c-Src [32]. PELP1 directly binds to c-Src with distinct domains, in which the N-terminal PXXP engages the Src SH3 domain, while residues 887–962 in the C-terminus bind the SH2 domain [10]. PELP1 binds integrin-linked kinase 1 (ILK1), and estrogen stimulation enhances this interaction within a signaling cascade involving ER, Src, and PI3K [33]. PELP1 is a substrate of CDKs (inculding cyclin-dependent kinase 2 (CDK2) and cyclin-dependent kinase 4 (CDK4)) and it is phosphorylated by CDKs in a cell cycle-dependent manner, which influences its localization and stability [15]. Several additional kinases including Protein kinase A (PKA), mechanistic target of rapamycin (mTOR), glycogen synthase kinase 3β (GSK3β), and DNA damage-induced kinases (ATM and ATR) phosphorylate PELP1, affecting its interactions and function [11,34] as shown in Figure 2.

Chromatin Modifiers: PELP1’s C-terminal glutamic acid-rich region (GAR) interacts with histones [29] and is involved in modifying chromatin and regulating gene expression. PELP1 interacts with SETDB1, a histone methyltransferase, enhancing SETDB1’s methyltransferase activity and promoting Akt activation [35]. PELP1 interacts with the histone variant macroH2A1 in MCF-7 human BCa cells, and their co-recruitment regulates gene expression [36]. PELP1 acts as a “reader” and uniquely binds to histones, acetyltransferases (CBP/p300), deacetylases (HDAC2), and the demethylase KDM1A/LSD1 [37,38,39]. By forming a complex with ERα and KDM1A, PELP1 can change the substrate specificity of KDM1A, facilitating the demethylation of H3K9 [40]. PELP1 binds to p53, altering p53’s phosphorylation and acetylation [1]. PELP1 and the arginine methyltransferase CARM1 interact to play a crucial role in arginine demethylation [37]. PELP1 interacts with arginine methyltransferase PRMT6 which is co-recruited to ER genes and regulates the RNA splicing [14]. PELP1 interacts with AIB1 (NCOA3)-containing complexes, and such interactions are shown to promote advanced cancer phenotypes [41].

Ribosome modifiers: Several of the PELP1 interactors are shown to play a critical role in ribosome biogenesis. PELP1 acts as a scaffold protein which forms a core complex with WDR18 and TEX10, which stabilizes the rixome and positions it on pre-60S ribosomes [6]. Additionally, cryo-EM studies provided architectural details of multi-protein complexes of PELP1 to bridge the enzymatic subunits of LAS1L, NOL9 (making RNase PNK complex), and SENP3 (which regulates SUMOylation of rixosome proteins) [6,42]. MDN1 (AAA-ATPase) binds to the C-terminus of PELP1, enabling regulation and mechanical functions during ribosome maturation [6,43]. The small linear motif (SLiM) exists in the intrinsically disordered regions (IDRs) which bind and activate SENP3 [6,44,45].

Collectively, these findings suggest that PELP1 functions as an oncogenic scaffold that integrates NR signaling, cytoplasmic kinase pathways, chromatin remodeling, and ribosome biogenesis, positioning it as a central regulator of transcriptional and translational programs in cancer.

## 3. PELP1 Signaling Pathways

### 3.1. Genomic Functions

PELP1 predominantly localizes to the nuclear compartment and functions as a coregulator of several NRs [1,31]. PELP1 also influences the gene expression from transcription factors AP1, SP1, NF-κB [29], STAT3 [30], and FHL2 [3] by functioning as a co-activator. Emerging evidence indicated that PELP1 achieves its genomic functions via its interactions with histones and chormatin-modifying enzymes [11]. PELP1 recruits E2F1 to pRb/E2F target gene promoters to alter its transactivation capabilities, activating genes involved in DNA replication and proliferation [15]. PELP1 facilitates E2-induced AR signaling by forming a protein complex with AR and ESR2 on the DNA, leading to the proliferation of prostate cancer (PCa) cells in the absence of androgen, allowing for crosstalk between these steroid receptors. PELP1 interacts with the promoters of miR-200a and miR-141, regulating their expression via engaging histone deacetylase 2 (HDAC2) [39]. PELP1 is associated with growth factor regulation of STAT3 activation, enhancing target genes including cyclin D1, c-myc, and c-fos [30]. PELP1 bridges with HER2 and regulates chromatin and increases aromatese expression [46].

### 3.2. Non-Genomic Signaling

PELP1 was initially identified as a Src SH2 domain-binding protein, where it reinforces ER-Src interactions and activates the Src-MAPK signaling pathway [3,47,48]. The AR and PELP1/MNAR/Src complex activates the Src-MEK-1/2-ERK-1/2-CREB pathway along with the androgen-independence phenotype [3,9]. ERα and ERβ interact with LXXLL motifs of PELP1 to bind with Src in the cytosol [10,48,49]. PELP1 serves as an essential scaffold and promotes non-genomic signaling by NRs. PELP1 links ESR1 to Src kinase in the cytoplasm, resulting in the activation of the ESR1–Src–MAPK pathway and PI3K–Akt pathway [33,50]. Estradiol stimulates MMP-9 production in ERα^+^ BCa cells via a PELP1-dependent, PI3K/Akt-mediated signaling pathway [51]. PELP1 facilitates epithelial–mesenchymal transition (EMT) and cell migration via the ER–Src–PELP1–ILK1 pathway [33]. PELP1 modulates Gβγ/AR-dependent non-genomic signaling that governs meiosis [52]. PELP1 is also implicated in alterations in AR non-genomic signaling during androgen-independence [9]. Hepatocyte growth factor-regulated tyrosine kinase substrate (HRS) interacts with PELP1, increasing MAPK pathway activation in an EGFR-dependent manner. The interaction between PELP1 and mTOR resulted in the activation of mTOR downstream signaling [50]. PELP1 serves as both an interaction partner and substrate of GSK3β, hence connecting PELP1 to Wnt/β-catenin and survival pathways [4]. Cytoplasmic PELP1 activates signaling pathways that converge on ERRγ to enhance cell survival [53]. PELP1 is a component of autophagosomes, linking estrogen signaling, autophagy, and cancer cell fate via HRS, suggesting new therapeutic avenues for hormone-responsive cancers [54].

### 3.3. Cell Cycle

PELP1 is a substrate of CDKs, specifically CDK2 and CDK4, which are active during the interphase of the cell cycle. Phosphorylation by CDK helps in regulation of the nuclear localization of PELP1 in cell cycle activation [15]. During the ribosomal transcription, PELP1 is found in the nucleolus during the S and G2 phases of the cell cycle [13]. PELP1 interacts with pRb in the C-terminal region, activating cyclin D1 and promoting cell cycle progression. PELP1 increases pRb hyperphosphorylation and enhances E2F1 transactivation [55]. PELP1 binds to pRb/E2F1 target promoters, linking estrogen signaling to cell cycle machinery [15]. PELP1 contributes to meiosis by interacting with Gβγ and the AR [52]. EGF can cause PELP1 phosphorylation at CDKs’ targeted sites, leading to PELP1 functions during cell cycle signaling [56].

### 3.4. Chromatin Modifications

PELP1 possesses a histone-binding domain that interacts with histones. PELP1 is involved in chromatin remodeling by displacing histone H1 in cancer cells [57]. PELP1 histone binding domain is also shown to recognize hypoacetylated histones H3 and H4, inhibiting their conversion into substrates for histone acetyltransferases [29]. Histone peptide array studies indicated that PELP1 also detects histones modified by dimethylation of arginine and lysine [37]. The histone variant macroH2A1 represses transcription, and PELP1 is shown to be a ligand-independent macrodomain-interacting factor, macroH2A1 and PELP1 also co-operate to regulate gene expression [36]. PELP1 interacts with histone demethylase KDM1A (LSD1) and alters histone methylation such as H3K4me2 and H3K9me2 [58]. PELP1 acts as a reader of histone methylation marks for ligand-bound ERα [40]. PELP1 is associated with arginine methyltransferase CARM1 in chromatin modulation, and PELP1 overexpression alters epigenetic states at ERα target promoters [37]. PELP1 interacts with chromatin modifier PRMT6 and its recognition of dimethyl-modified histones, leading to alternative splicing outcomes [14]. PELP1 via the Rix1 complex plays a role in heterochromatin maintenance, involved in heterochromatin RNA degradation/processing [6]. PELP1, the fundamental component of 5FMC, links arginine methylation and (de)SUMOylation to transcriptional activity [59]. Polyglutamylase TTLL4 glutaminates PELP1, which coordinates chromatin remodeling by PELP1 [20]. Collectively, these findings indicate that PELP1 serves as a versatile chromatin regulator in cancer cells by recognizing histone modifications and via interactions with chromatin-modifying enzymes to influence transcription.

### 3.5. DNA Damage Response

PELP1 is a novel substrate of the DDR kinases ATM, ATR, and DNA-PKcs. PELP1 is phosphorylated at the Serine 1033 residue in the C-terminus in response to genotoxic stress [22]. PELP1 interacts with p53, and the phosphorylation of PELP1 by DDR kinases is crucial for the coactivation functions of p53. PELP1-depleted p53 (wild-type) BCa cells were less susceptible to a variety of genotoxic agents, such as gamma radiation, camptothecin, or etoposide [22]. PELP1 interacts with mutant p53, regulates its recruitment, and alters epigenetic marks at the target gene promoters [17]. PELP1 deletion reduces E2F target activation and impairs DNA repair in triple-negative breast cancer (TNBC) cells, making them more susceptible to DNA-damaging agents [17]. Collectively, these results identify PELP1 as an essential mediator of DNA damage response signaling, functioning as a substrate for DDR kinases and a coactivator for p53 and E2F transcriptional pathways to enhance cancer cell viability during genotoxic stress. These findings demonstrate that PELP1 serves as a crucial sensor and regulator of DNA damage response signaling through its phosphorylation by DDR kinases and its role as a coregulator of the p53 and E2F pathways.

### 3.6. Immune Signaling

PELP1 functions as a coregulator of NF-κB and is also implicated in the modulation of inflammatory signals leading to macrophage activation [11,29,60]. PELP1 also functions as a coregulator of STAT3, a transcription factor that plays a essential role in immune signaling [30]. PELP1 is an inflammation-inducible gene, and its expression is induced by NF-κB subunit c-Rel [61]. LPS/c-Rel drives PELP1 expression in macrophages; PELP1 overexpression triggers granulocyte–macrophage colony-stimulating factor (GM-CSF) secretion, resulting in tumor progression via the microenvironment [61]. PELP1 KD decrease NF-κB activity in medullbalstoma and blocks activation of p65 [62]. PELP1 cytoplamic signaling is shown to increase NF-κB expression in BCa, resulting in activation of inflammatory cytokines and chemokines. Cytoplasmic PELP1 promotes the migration of breast epithelial cells by upregulating the inhibitor of kappaB kinase and facilitating inflammatory interactions with macrophages [63]. Breast cancers with a migratory character are caused by PELP1 being localized in the cytoplasm, which increases pro-tumorigenic IKK and NF-κB signaling [63]. High PELP1 is associated with elevating T regulatory (Treg) cell presence and has a negative correlation with immune checkpoints PD-L1 (CD274) and CTLA4 [64]. These findings suggest that PELP1 has potential to modulate tumor micro environment via modulation of NF-κB and STAT3 signaling.

### 3.7. Stem Cells

PELP1 signaling plays a role in the biology of stem cells. PELP1 interacts with AIB1 to regulate stemness by increasing Thr24 phosphorylation, and PELP1/AIB1 axis signaling supports stemness by self-renewal [41]. Cytoplasmic PELP1 expression is associated with increasing the number of ALDH tumor spheres in a BCa model [41]. PELP1 expression is upregulated during the osteogenic differentiation of human periodontal ligament stem cells (hPDLSCs) [65], and the PELP1/RUNX2 axis plays a role in stem cell fate [65].

### 3.8. Ribosome

The PELP1-TEX10-WDR18 complex functions as a ribosome biogenesis regulator, and PELP1 localization regulates the rate of ribosome synthesis, leading to cancer progression [18,66]. PELP1 localizes to the nucleolus and plays a role in optimal production of the 28S rRNA [67]. Through the controlled recruitment and release of the AAA ATPase MDN1, PELP1 has been demonstrated to act as a regulatory point for mammalian 60S maturation [43]. The structure of the human PELP1-Rix1 domain coupled to WDR18 was established in recent research utilizing cryo-electron microscopy (cryo-EM) at a resolution of 2.7 Å [42]. Based on the newly released cryo-EM structure of the WDR18/PELP1-Rix1 complex (PDB code 7UWF) [42], the PELP1 homodimer acts at the core of the WDR18/PELP1 assembly. By being recruited to rDNA promoters and enhancing rRNA transcription, PELP1 demonstrates dual regulatory roles in ribosome biogenesis [13,66]. Through interactions with PELP1, TEX10, WDR18, NOL9, and SENP3, LAS1L contributes to the formation of a nucleolar complex, homologous to the yeast Rix1 complex, that cofractionates with the 60S preribosomal subunit [67]. In addition, rixosomal RNA degradation contributes to Polycomb target gene silencing (contextualizes PELP1’s complex in chromatin repression) [68]. SENP3/USP7 regulate Polycomb–rixosome coupling and list PELP1 as a rixosome subunit in human cells [69]. PELP1 sits in the core scaffold of the human rixosome (PELP1–WDR18–TEX10–LAS1L), organizing subunit assembly for both ribosome biogenesis and Polycomb-target silencing [70]. Cryo-EM showed that PELP1 interacts with WDR18 and TEX10 to form a platform that directs the LAS1L–NOL9 catalytic module [70]. PELP1 acts as a scaffold protein, a structural organizer, and an allosteric gatekeeper, coupling rRNA processing machinery to chromatin-silencing functions [70].

## 4. Role of PELP1 in Cancer

### 4.1. Breast Cancer (BCa)

PELP1 is expressed in BCa cells, and its expression is upregulated in breast tumors [12,71] (Figure 3). PELP1 tumor levels are correlated with estogen levels in normal tissues and plasma in postmenopausal women [72]. PELP1 translocates between the nucleus and cytoplasm, but is primarily nuclear in normal breast tissue [41]. PELP1 contains proximal estrogen response element (ERE) half-sites within its promoter, as it acts as an estrogen target gene [73]. Additionally, PELP1 is shown to regulate expression of the aromatase gene and it has potential to contribute to in situ estrogen synthesis in tumors [27]. PELP1 expression correlated with E1 and E2 concentrations in plasma, normal tissue, and tumor samples, while no correlations were observed in ER-negative tumors [72]. PELP1 upregulation is linked to endocrine therapy resistance in BCa [72]. PELP1 regulates the epigenetic changes on ER target gene promoters through its interactions with lysine-specific histone demethylase 1 (KDM1) [58]. PELP1 knockdown (KD) reduces cancer proliferation and therapy resistance by activation of KDM1 [58]. PELP1 overespression increases sensitivity to the cells to S-phase entry via the regulation of pRb phosphorylation and cyclin D1 [55]. PELP1 is shown to promote BCa progression via its interactions with STAT3, an oncogene highly expressed in BCa’s [30].

Interestingly, engineered clones of BCa cells expressing cytoplasmic PELP1 are tumorigenic in vivo, promote apototic sensitivity changes, and are estrogen-independent, highlighting the role of PELP1 subcellular localization in cancer progression [74]. Cytoplasmic PELP1 model cells are hypersenesitive and promote tamoxifen and paclitaxel resistance [32,75]. For instance, RNA-seq analysis showed upregulated genes such as ABCB1 and TUBB2B in the paclitaxel-resistant MCF7 tumorsphere [75]. Multi-omics analyses support the role of cytoplasmic and nuclear PELP1/SRC-3 complexes (enriched in ER^+^) in reprogamming the transcriptional and epigenomic landscape of resistant tumors [75]. Additionally, CUT&RUN analyses illustrate the overlapping of PELP1/SRC-3 complexes in chromatin-binding peaks with pathways such as androgen response, cholesterol homeostasis, Notch, and TGF-β signaling [75].

Cytoplasmic PELP1 is shown to increase tamoxifen resistance in BCa. PELP1-cyto and ERRγ protect HMECs from tamoxifen-induced death, highlighting PELP1 localization as a predictive biomarker of tamoxifen response [53]. Liu et al. showed that PELP1 is a novel interactor with SETDB1, inducing endocrine resistance in ER^+^ BCa [35], and PELP1 KD reduces SETDB1-AKT methylation, phosphorylation, and ER signaling [35]. RNA-seq analyses showed that SETDB1/PELP1-regulated genes overlap, with respect to ER signaling, tamoxifen resistance, and PI3K–Akt signaling, suggesting that cytoplasmic localization of PELP1 in breast tumors may enhance SETDB1’s oncogenic functions [35]. PELP1 interactions with SRC-3 promote the expansion of BCa stem cells by upregulating HIF-driven metabolic genes including PFKFB3 and PFKFB4 [76]. Further, PELP1 is known to promote both glycolysis and mitochondrial respiration by engaging with PFKFB3/4, thereby driving tumorsphere formation and CTC emergence, making PELP1 contribute to metastatic progression and therapy resistance in patient-derived organoid models [76].

**Figure 3 biomolecules-15-01729-f003:**
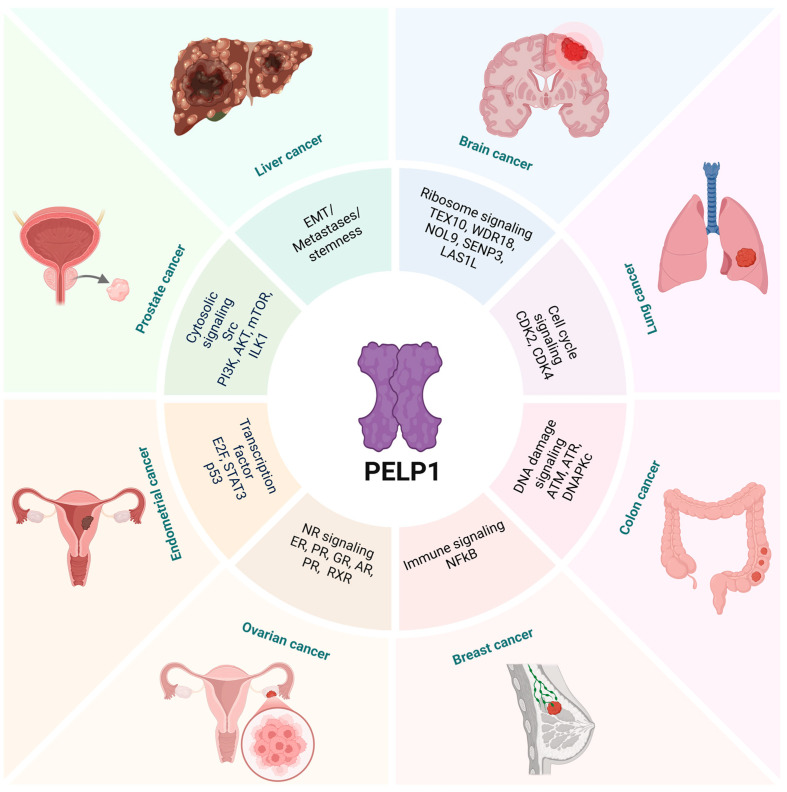
PELP1 plays a critical role in the progression of multiple cancers. This schematic shows different signaling pathways and biological processes that PELP1 controls in cancer cells. PELP1 serves as a scaffolding oncoprotein, coordinating nuclear receptor signaling, transcriptional regulation, ribosome biogenesis, DNA damage response, and cytoplasmic kinase cascades. Emerging finding suggest that PELP1 acts as a central hub that brings together nuclear and cytoplasmic signaling networks that promote oncogenic transcription, survival signaling, and metastasis in many types of cancer.

Unbiased global analyses using PELP1 knockout TNBC cells suggested that PELP1-mediated oncogenic functions involve regulation of transcription factors such as c-Myc and ribosomal biogenesis in TNBC [77]. PELP1 modulates genes that are involved in the epithelial–mesenchymal transition (EMT), including MMPs, SNAIL (SNAI1), TWIST (TWIST1), ZEB (ZEB1), MYC, MTA1, miR-200a, and miR-141, leading to cancer metastasis progression [33,39,78,79]. PELP1 is shown to improve the effectivness of chemotherapy in TNBC by blocking cell cycle regulation and promoting apoptosis [17]. PELP1 interacts with MTp53, guiding its promoter recruitment, and modifies epigenetic regulation at NFY target gene promoters through the histone modifications at this site [17,80]. PELP1 KD reduces MTp53, leading to reducing cell viability, and increases apoptosis by regulating E2F1 gene expression [17]. PELP1 controls the stability of E2F1 in opening chromatin (regulator of cell cycle and apoptosis) through a mechanism dependent on KDM1A demethylation [17] (PELP1 interacts with KDM1A [40]). PELP1 is a substrate of ATM in WTp53-mediated apoptosis in ER^+^ BCa [22]. PELP1 phosphorylation at Ser1033 mediated by DNA damage kinases increases in TNBC subtypes, more than other BCa subtypes and normal tissues [17]. Another study showed that PELP1 is increased in an HIF-dependent manner in TNBC cells by interacting with the glucocorticoid receptor (GR) to activate Brk (breast tumor kinase) expression [81]. Collectively, these findings suggest that PELP1 oncogenic signaling faciliates progression of both ER^+^ and TNBC subtypes of BCa.

### 4.2. Prostate Cancer (PCa)

PELP1 plays a pivotal role in PCa progression by binding to FHL2 [3]. PELP1 binds to FHL2 in vitro and in vivo, co-localizes in the nucleus, and together synergistically enhances FHL2 reporter genes in PCa [3]. Src kinase is needed for PELP1/FHL2 activation, and inhibition of Src reduces both PELP1-dependent and synergistic PELP1–FHL2 transactivation [3]. PELP1 bridges ERβ and AR in PCa, enabling estrogen to activate AR signaling through PELP1 LXXLL motifs even in low AR conditions [82].

### 4.3. Endometrial Cancer (ECa)

The E2-mediated cell proliferation of ECa cells is significantly influenced by PELP1, which is associated with the interaction between pRb and E2-mediated cell cycle progression [23,55]. In the postmenopausal endometrium, PELP1 localization is dynamic, and it localizes in the cytoplasm, in parallel with ERβ [23]. However, it shifts to the nucleus during the proliferative and secretory phases of the premenopausal endometrium [23,71]. PELP1 interacts with ERβ in the cytoplasm, forming a complex that may contribute to tumorigenesis via nongenomic pathways, including activation of Src-MAPK signaling [23]. PELP1 also may enhance tamoxifen’s partial agonist effects in ECa cells but not in BCa cells, implicating the role of PELP1 in the tissue-specific actions of selective ER modulators [23,83,84]. PELP1 is implicated in the enhanced invasion and migration of endometrial cancer cells induced by tamoxifen and fulvestrant through a Src-dependent mechanism [23].

### 4.4. Ovarian Cancer (OCa)

PELP1 expression is elevated by 2-3 fold in 60% of ovarian tumors and is involved in the progression of OCa [85]. PELP1 overexpression contributes to the activation of c-Src kinase and AKT, and it forms complexes with Src and the p85 subunit of PI3K in OCa cells [85]. PELP1 KD induces alterations in actin reorganization and alters the expression of metastasis-related genes in OCa cells [86]. PELP1 KD reduces OCa cell growth in vitro and tumor progression in vivo [86]. PELP1 promotes angiogenesis in epithelial ovarian cancer (EOC) cells through the regulation of VEGFA expression and secretion [87]. Further, PELP1 upregulation increases proliferation, migration, invasion, and metastasis of EOC cells by inducing expression of EMT markers such as N-cadherin and Vimentin [87]. PELP1 interactions with ESR1 and ESR2 and tyrosine kinase c-Src are shown to contribute to OCa progression [88].

### 4.5. Hepatocellular Carcinoma (HCC)

PELP1 expression is higher in HCC than in normal tissue. PELP1 KD decreases cell viability, colony formation, and invasion in HCC cells and induces apoptosis. The growth of HCC tumors in xenograft models was substantially slowed by the knockdown or pharmacologic inhibition of PELP1 [8].

### 4.6. Colorectal Cancer (CRC)

PELP1 signaling contributes to CRC progression by enhancing cell viability, clonogenicity, invasion, and tumor progression. PELP1 KD reduces CRC via c-Src downregulation [21]. PELP1 signaling may also occur independent of c-Src via GR [21]. Additionally, PELP1 drives angiogenesis via the STAT3/VEGFA axis in endothelial cells, PELP1 KD significantly reducing tumor growth and angiogenesis in vivo [89].

### 4.7. Non-Small Cell Lung Carcinoma (NSCLC)

In NSCLC, PELP1 is overexpressed, which exacerbates tumor cell malignancy and resistance to tyrosine kinase inhibitor drugs. PELP1 KD is shown to reduce cell viability, clonogenicity, migration, and invasion [64]. Further, PELP1 KD induces sensitivity to the EGFR tyrosine kinase inhibitor (TKI) gefitinib [64]. RNA-seq studies showed that PELP1 signaling modulate pathways involved in EGFR TKI resistance, MAPK, PPAR, metabolism, autophagy, cytokine signaling, and oncogenic targets included RET, MAPK4, and NRG2 (linked to TKI resistance) [64].

### 4.8. Gastric Cancer (GCa)

PELP1 KD decreases cell viability, clonogenicity, migration, and invasion of GCa xenograft models, and PELP1 overexpression increases proliferation, colony formation, and migration in vitro [90]. Knockdown of PELP1 decreases activity of the c-Src–PI3K–ERK pathway and regulates transcription of PI3K and ERK, suggesting that PELP1 drives GCa progression through this signaling axis [90]. PELP1’s oncogenic role in GCa appears independent of ERα/ERβ, which are expressed at low levels in gastric tumors [90].

### 4.9. Medulloblastoma (MB)

PELP1 signaling plays an essential role in the progression of MB. PELP1 KD decreases NF-κB target genes, and IKKα/β–IκB signaling, and p65 activation.. Additionally, PELP1 KD reduces MB cell viability, proliferation, survival, and invasion. RNA-seq analyses of PELP1 KD cells showed that PELP1 affected cell adhesion, inflammation, extracellular matrix remodeling, invasion, and angiogenesis [62].

### 4.10. Pancreatic Cancer

Polyglutamylation of PELP1 in pancreatic cancer may influence its affinity for interacting proteins like LAS1L and SENP3 and the composition or conformation of the MLL1-WDR5 supercomplex, affecting histone modification and chromatin structure. TTLL4 is commonly overexpressed in pancreatic cancer, and PELP1 polyglutamylation by TTLL4 may affect the composition or conformation of highly organized chromatin remodeling complexes like the MLL1-WDR5 complex and contribute to pancreatic cancer cell proliferation through histone modifications and chromatin remodeling [20].

## 5. PELP1 as a Biomarker

PELP1 is highly expressed in invasive BCa patients’ tissues, demonstrating large-sized tumors, higher mitotic count, and elevated histological grade [91]. PELP1/Ki-67 high expression in tumors is an independent prognostic factor for predicting poor survivorship in TNBC patients [92]. PELP1 is upregulated in non-luminal BCa and is associated with luminal B BCa [93]. PELP1 is highly expressed in BCa patients specifically in stage 3, and PELP1 is high in metastatic TNBC [94,95,96]. The cytoplasmic localization of PELP1 was seen in 36% of women at elevated risk for BCa development [97]. PELP1 is high in breast fibroadenoma, and an ER/HER-2-positive group had significantly higher PELP1 H-scores than their negative group [98].

PELP1 promotes tumor progression and metastasis, particularly in non-luminal subtypes (better than GATA3), and is associated with poor survival outcome [91,93]. Higher PELP1 protein expression is associated with various features, including higher tumor grade, increased proliferation, node-positive invasive disease, distant metastases, and reduced BCa-specific factors as well as disease-free survival [2,91]. PELP1 is high in NSCLC, with positive lymph node metastasis, elevation of ERα, ERβ, and PCNA [99]. PELP1 (mRNA and protein levels) is high in CRC compared to immortalized normal colorectal epithelium [21]. PELP1 expression is positively correlated with advanced tumor stage and lymph node metastasis in GCa [90].

In ECa, PELP1 expression is significantly higher in ECa tumor tissues compared to normal tissues [100]. In OCa, positive PELP1 expression was associated with better disease-free survival and overall survival [101]. PELP1 expression is found to be dysregulated across all major subtypes of OCa including serous, endometrioid, clear cell, and mucinous carcinomas [71,85]. PELP1 and VEGFA exhibit a robust positive correlation in OCa, and both are elevated in OCa [87]. Conversely, in some OCa subsets, PELP1 expression, particularly alongside Erβ, may indicate a more differentiated phenotype and improved prognosis [101]. This underscores the need for context-dependent interpretation of PELP1 as a biomarker and highlights the complexity of its role across cancer types.

PELP1 is dysregulated in 80–95% of PCa cases, especially in advanced and castration-resistant disease, associated with shorter survival rate and enrichment of pathways related to PCa progression, endocrine resistance, and hormone signaling [102,103]. Additionally, PELP1 is a diagnostic and prognostic biomarker in metastatic castration-resistant PCa [103]. Increased expression of PELP1 relative to normal tissues was reported in HCC, and higher PELP1 expression correlates with poor survival [8]. PELP1 expression is high in MB tissues compared to normal brain tissue [62]. PELP1 is high in GCa cell lines and patient tumor samples compared to normal cells [90]. PELP1 and ERβ are co-localized in various cell types within CRC tumors, while ERβ expression correlates with improved outcomes, and PELP1 overexpression in epithelial cells is an independent favorable prognostic factor [104].

High PELP1 expression is linked with greater invasion depth, lymph node metastasis, higher tumor histological grade, and advanced TNM stage, but not with age, sex, tumor size, or tumor number in lung cancer [90]. PELP1 expression is high in NSCLC samples [64,105]. PELP1 is associated with squamous histology, smoking, and wild-type EGFR status [64]. Collectively, these findings support the utility of PELP1 as a biomarker for cancer prognosis.

## 6. Therapeutic Targeting of PELP1

PELP1 serves as a scaffolding protein devoid of any known enzymatic activity. Consequently, numerous studies have investigated targeting of PELP1 by siRNA as well as focused on PELP1 downstream pathways or disrupting PELP1 dimerization (Figure 4). PELP1 siRNA-loaded liposomes were efficient in reducing tumor growth of BCa [58] and OCa [86]. KDM1A pharmacological inhibitors also decreased PELP1-mediated growth of BCa tumors [58]. The antipsychotic drug chlorpromazine (CPZ) inhibits GCa aggressiveness by downregulating PELP1 expression [90]. Metformin induces pyroptosis in esophageal squamous cell carcinoma (ESCC) by modulating the miR-49F7/PELP1 signaling axis [106]. The CDK2 inhibitor roscovitine reduces the expression of PELP1, suggesting that CDK2 inhibition has the potential to block the proliferation of PELP1-driven BCa [107]. Rapamycin and AZD8055 reduced proliferation of PELP1-overexpressed BCa cells, suggesting that mTOR inhibitor(s) are effective in blocking PELP1 oncogenic functions [50]. PR antagonists are shown to have therapeutic utility in treating ESR1 and PR^+^ BCa by blocking functions of ESR1-PELP1-IGF1R-containing complexes [25]. PELP1 is shown to interact with NRs via LXXLL motifs. Peptidomimetic D2 (LXXLL mimetic) blocks the interaction between AR and PELP1 and blocks DHT-induced AR nuclear translocation and genomic activity [24].

Our laboratory recently identified SMIP34 as a novel and first-in-class PELP1 inhibitor [108]. MST assay shows direct binding for SMIP34 to PELP1 in a dose-dependent manner. Mechanistically, SMIP34 binds to PELP1 and destabilizes the dimer and degrades the proteasome [108]. Based on the cryo-EM structure of the WDR18/PELP1 Rix1 complex (PDB: 7UWF), PELP1 dimerization mediated by α-helices 17, 20, and 22 is essential for complex formation. Published data [108] and AlphaFold predictions indicate that SMIP34 binds to the loop region (aa 696–720) near the dimer interface, disrupting homodimerization and WDR18/PELP1 assembly, leading to PELP1 destabilization and degradation. Subsequent studies showed that SMIP34 binds to the PELP1 aa 696–720 loop, attenuating PELP1 homodimerization and preventing the assembly of the WDR18/PELP1 Rix1 complex [7]. SMIP34 reduced proliferation and tumor growth in both wildtype and MT-ER^+^ BCa models [108]. Pharmacological inhibition of PELP1 using SMIP34 significantly inhibited the growth of HCC tumors [8]. Collectively, these findings suggest that either directly targeting PELP1 or its signaling complexes will have utility in treating PELP1 deregulated cancers. PELP1’s unique scaffold roles, including integrating chromatin remodeling, transcriptional regulation, and ribosome biogenesis, make it a distinct and superior therapeutic target compared to proteins involved in only one or two processes.

## 7. Conclusions and Future Directions

As a central scaffolding hub, PELP1 orchestrates critical events of oncogenic signaling by converging interactions between nuclear receptor signaling, chromatin modifications, cell cycle transitions, and cytoplasmic kinase pathways. In addition, PELP1’s distinct localization, widespread post-translational modifications, and ability to form discrete signaling complexes contribute to its diverse effects on cancer cell proliferation, survival, and therapeutic resistance. Recent cryo-EM structural studies discovered a new role of the PELP1-associated Rix complex in translational control and chromatin modification, providing additional avenues to target the PELP1 axis. Current cryo-EM studies have only solved PELP1’s N- and C-termini. Further research on PELP1’s full-length structure will help explain its biological roles. Considering the emerging role of PELP1 in DNA damage response, investigating how PELP1 contributes to DNA damage repair and replication stress will be beneficial. Also, PELP1’s impact on the TME concerning immunology, cytokine pathways, and signaling cascades will elucidate additional signaling pathways of therapeutic potential to block PELP1 oncogenic functions.

Although significant progress has been made in understanding PELP1 biology and developing strategies to target its oncogenic functions, several limitations remain. PELP1 lacks intrinsic enzymatic activity, making it a challenging target for conventional small-molecule drug design. Current approaches, such as siRNA delivery, peptidomimetics, and indirect targeting through downstream signaling pathways, face hurdles related to stability, specificity, and efficient delivery to tumor sites. The identification of the first-in-class PELP1 inhibitor confirms the druggability of PELP1 despite its scaffolding nature and opens avenues for developing more potent nM-range inhibitors for human translation.

Future efforts should focus on improving the pharmacokinetic properties and tumor selectivity of PELP1 inhibitors, such as SMIP34, and exploring combination strategies with endocrine therapy, CDK inhibitors, DNA damage response, or immunotherapy to overcome resistance mechanisms. Development of robust biomarkers to identify patients with PELP1-driven tumors will be critical for patient stratification. Additionally, addressing potential off-target effects and toxicity profiles will be essential for advancing PELP1-targeting agents into clinical trials. Overall, translating these promising preclinical findings into effective therapies will require multidisciplinary approaches integrating medicinal chemistry, drug delivery systems, and precision oncology.

## Figures and Tables

**Figure 1 biomolecules-15-01729-f001:**
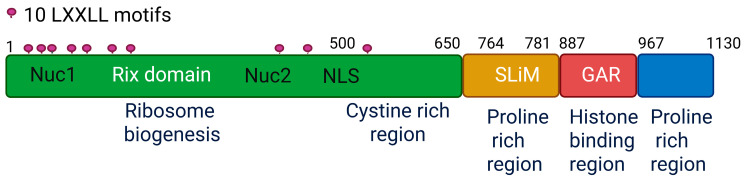
Structural composition of the human PELP1 protein and its functional domains. Schematic depiction of PELP1 (amino acids 1–1130) highlighting its principal domains and motifs. The N-terminal region (1–650 aa) encompasses the Rix domain, essential for ribosome synthesis, and contains several LXXLL motifs (totaling 10) that facilitate interactions with nuclear receptors. This area contains a nuclear localization signal (NLS) that enables nuclear transport and two nucleolar localization motifs, Nuc 1 (79–160 aa) and Nuc 2 (423–489 aa), which help localize PELP1 in the nucleolus. The cysteine-rich region (500–650 aa) coincides with the Rix domain. The central segment (650–887 aa) contains a brief linear motif (SLiM; 764–781 aa) that facilitate SENP3 interaction. The GAR (glutamic acid rich) domain (887–966 aa) is associated with histone binding and chromatin remodeling. The C-terminal segment (967–1130 aa) has a second proline-rich region, which is linked to scaffolding functions.

**Figure 2 biomolecules-15-01729-f002:**
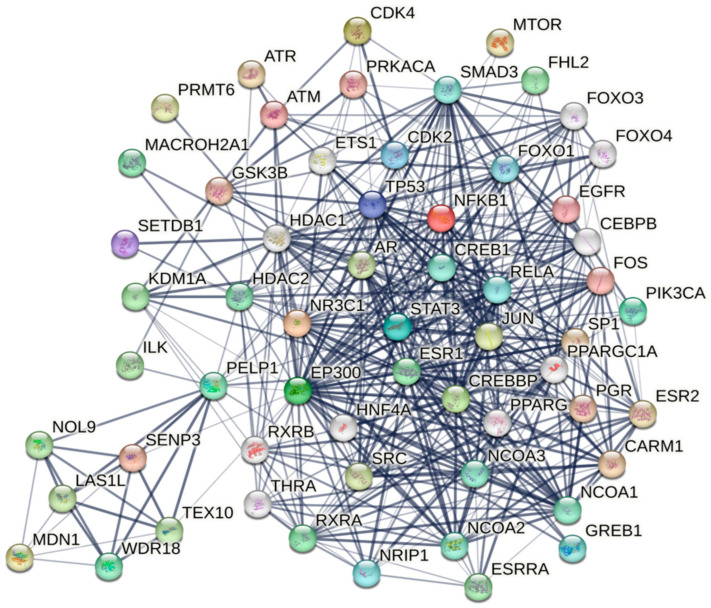
PELP1 interaction network and its associated partners. The figure illustrates experimentally validated and highly probable predicted protein partners of PELP1, derived from the STRING database (interaction confidence ≥ 0.7). PELP1 is the central molecule, linking multiple functional clusters vital for NR signaling, chromatin remodeling, kinase signaling, DNA damage response, and cell cycle regulation. Nuclear receptors (ERα/ESR1, ERβ/ESR2, AR, PR, GR/NR3C1, RXRα, RXRβ, and PPARG) and chromatin modifiers (EP300, CREBBP, NCOA1–3, KDM1A/LSD1, SETDB1, HDAC1/2, PRMT6, CARM1, MACROH2A1, and SENP3), along with transcription factors (STAT3, NFκB1, CREB1, SP1, TP53, and HNF4A), interact with PELP1. The PELP1 network also encompasses components involved in ribosome biogenesis (MDN1, TEX10, WDR18, LAS1L, NOL9) as well as kinases (SRC, EGFR, CDK2/4, AKT/GSK3β, and MTOR). These interactions underscore PELP1's scaffolding function in oncogenic signaling.

**Figure 4 biomolecules-15-01729-f004:**
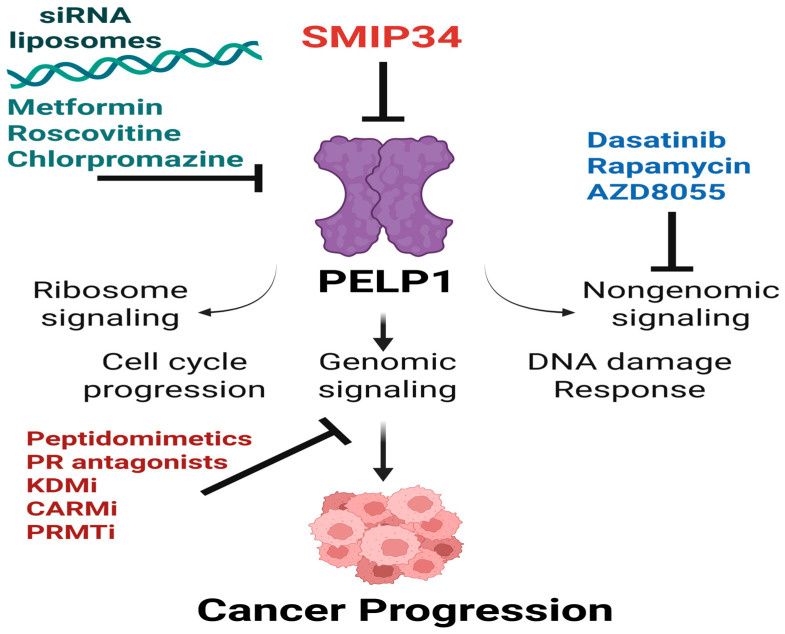
A schematic view of multiple pharmacological and molecular techniques designed to inhibit PELP1 oncogenic signaling pathways. Small molecules like SMIP34 directly target PELP1, destabilizing and promoting its degradation. Gene silencing by PELP1 siRNA-loaded liposomes significantly reduces tumor growth in preclinical models. Further, indirect suppression of PELP1 expression by repurposed drugs such as metformin, roscovitine, and chlorpromazine is also shown to reduce PELP1 effects in cancer cells. Non-genomic PELP1 signaling can be inhibited by Dasatinib, Rapamycin, and AZD8055, targeting PELP1-associated cytoplasmic kinases as well as mTOR pathways. Peptidomimetics, progesterone receptor (PR) antagonists, and epigenetic enzyme inhibitors (KDMi, CARMi, and PRMTi) are effective in blocking PELP1 genomic functions by controlling PELP1-driven transcription and chromatin regulatory actions. Overall, these approaches establish PELP1 as an important therapeutic target and support the concept of PELP1 targeting for treating solid cancers.

**Table 1 biomolecules-15-01729-t001:** Post-translational modifications of PELP1.

Modification	Site	Enzyme/ Regulator	Functional Outcome	Reference
Phosphorylation	Ser477, Ser991	CDKs	Enhances E2F1 activation, ER signaling	[15]
Phosphorylation	Ser350, Ser415, Ser613	PKA (Growth factor signals)	Enhances co-activation, nuclear redistribution	[16]
Phosphorylation	Ser1033	ATM, ATR, DNA-PKcs	DDR regulation, apoptosis regulation	[22]
Phosphorylation	Thr745, Ser1059	GSK3β	Regulates PELP1 stability	[4]
SUMOylation	Not specified	SUMO (SENP3-regulated)	Regulates ribosome maturation and trafficking	[18]
Ubiquitination	Lys496	Vps11/18	Prevents c-Src interaction, blocks ERα phosphorylation	[19]
Polyglutamylation	Not specified	TTLL4	Affects histone H3 affinity, chromatin remodeling	[20]
Phosphorylation	Tyr920	c-Src	Reciprocal regulation with Src	[21]

**Table 2 biomolecules-15-01729-t002:** PELP1 interacting proteins and their functional categories.

Category	Interacting Proteins	Functional Role	References
Nuclear receptors (NRs)	ERα, ERβ, AR, PR, GR, ERRα, RXR	PELP1 acts as a co-activator for NRs via LXXLL motifs, facilitating transcriptional activation.	[12,23,24,25,26,27,28]
Transcription factors/coregulators	AP1, SP1, NF-κB, STAT3, FHL2	PELP1 functions as a coregulator, modulating transcription factor-mediated signaling pathways.	[3,29,30]
Kinases and signaling proteins	c-Src, EGFR, PI3K, ILK1, CDK2, CDK4, PKA, mTOR, GSK3β, ATM, ATR	PELP1 acts as a scaffolding protein for kinases and is also phosphorylated by many kinases, affecting their localization, stability, and oncogenic signaling.	[10,11,15]
Chromatin modifiers	SETDB1, macroH2A1, CBP/p300, HDAC2, KDM1A/LSD1, p53, CARM1, PRMT6	PELP1 binds histones and epigenetic enzymes and regulates chromatin structure and transcription by interacting with histone methyltransferases/demethylases and acetyltransferases.	[1,14,29,31,32,33,34,35,36]
Ribosome biogenesis complex	WDR18, TEX10, LAS1L, NOL9, SENP3, MDN1	PELP1 is a component of the rixosome complex, critical for pre-60S ribosomal subunit maturation. Bridges enzymatic subunits and regulates SUMOylation and ATPase-driven maturation.	[5,6,37,38,39]

## Data Availability

No new data were created or analyzed in this study.

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
