# Peer review of "The PELP1 Pathway and Its Importance in Cancer Treatment"

_biomolecules, 2025, doi:10.3390/biom15121729_

Round 1

Reviewer 1 Report

Comments and Suggestions for Authors In this review article, Khaled et al.  provided a detailed summary of PELP1 biology, with focus on its diverse functions in cancer progression. The paper is thorough and included post-translational modifications, protein interactions, and recent findings from cryo-electron microscopy studies. The authors also clearly summarized the emerging findings regarding PELP1’s role in therapeutic resistance, immune modulation, and stemness. The review clearly established the clinical significance of targeting PELP1 and presents the novel idea that a scaffolding protein can be targeted by a small molecule, marking a significant conceptual advancement. Overall, this is a well-organized and timely review.  However, addressing the following minor concerns will strengthen the review article. Adding a table including post-translational modifications ( such as phosphorylation  sites) on PELP1 could be useful The review could be improved by adding limitations in current PELP1-targeting strategies and challenges for clinical translation in the future directions section. It can be accepted with those revisions ( Minor)

Author Response

Coment: Adding a table including post-translational modifications (such as phosphorylation sites) on PELP1 could be useful

Response: We thank the reviewer for this helpful suggestion. We have now included a comprehensive table summarizing the known post-translational modifications of PELP1, including phosphorylation sites, acetylation, SUMOylation along with their functional implications and associated references.

Coment: The review could be improved by adding limitations in current PELP1-targeting strategies and challenges for clinical translation in the future directions section. It can be accepted with those revisions (Minor)

Response: We thank the reviewer for this valuable suggestion. We have revised the Future Directions section to include the limitations of current PELP1-targeting strategies and the challenges associated with clinical translation. Specifically, we discuss the lack of enzymatic activity in PELP1, which complicates direct drug design, the reliance on indirect approaches such as siRNA or pathway inhibition, and the need for improved delivery systems and specificity to minimize off-target effects. Additionally, we highlight challenges such as pharmacokinetics, toxicity, and the requirement for robust biomarkers to guide patient selection in clinical trials.

Reviewer 2 Report

Comments and Suggestions for Authors

This manuscript, which provides a comprehensive and timely summary of proline, glutamic acid, and leucine-rich protein 1 (PELP1), is well structured and focuses on the emerging oncogenic functions of PELP1 and its potential as a therapeutic target for cancer treatment. This review systematically consolidates its various roles, which is valuable for researchers in oncology and molecular biology. The authors effectively address the multifaceted roles of PELP1 as an oncogenic scaffold.

The authors should provide more details or a structural model of how SMIP34 binds to and destabilizes the PELP1 dimer; this would enhance the translational impact of this topic. The conclusion should explicitly reiterate how this unique scaffold function makes PELP1 a superior or distinct therapeutic target compared to proteins involved in only one or two of these processes. Furthermore, the manuscript notes that PELP1 serves as a prognostic factor for poor survival in several cancers, but also that its expression has been associated with improved disease-free survival in some ovarian cancer (OCa) studies. A brief discussion reconciling these seemingly contradictory findings or highlighting the tissue-specific context would be helpful.

Author Response

Coment: The authors should provide more details or a structural model of how SMIP34 binds to and destabilizes the PELP1 dimer; this would enhance the translational impact of this topic. The conclusion should explicitly reiterate how this unique scaffold function makes PELP1 a superior or distinct therapeutic target compared to proteins involved in only one or two of these processes.

Response: Thank you for this suggestion. We have expanded the discussion to include structural insights into SMIP34’s mechanism. Based on the Cryo-EM structure of the WDR18/PELP1 Rix1 complex (PDB: 7UWF), PELP1 dimerization mediated by α-helices 17, 20, and 22 is essential for complex formation. Published data and AlphaFold predictions indicate that SMIP34 binds to the loop region (aa 696–720) near the dimer interface, disrupting homodimerization and WDR18/PELP1 assembly, leading to PELP1 destabilization and degradation. We also revised the conclusion to emphasize that PELP1’s unique scaffold role including integrating chromatin remodeling, transcriptional regulation, and ribosome biogenesis makes it a distinct and superior therapeutic target compared to proteins involved in only one or two processes.

Coment: Furthermore, the manuscript notes that PELP1 serves as a prognostic factor for poor survival in several cancers, but also that its expression has been associated with improved disease-free survival in some ovarian cancer (OCa) studies. A brief discussion reconciling these seemingly contradictory findings or highlighting the tissue-specific context would be helpful.

Response: We appreciate this insightful comment. We have now added a brief discussion in the manuscript to reconcile these observations. Specifically, we note that while PELP1 overexpression generally correlates with poor prognosis in several cancers, including breast and lung, some studies in ovarian cancer have reported improved disease-free survival when PELP1 expression is associated with ERβ positivity. These findings suggest that the prognostic role of PELP1 is tissue-specific and may depend on the hormonal receptor context and tumor biology.